# Impact of Carbon Tax and Environmental Regulation on Inbound Cross-Border Mergers and Acquisitions Volume: An Evidence from India

Chandrika Raghavendra [1,*], Mahesh Rampilla [1], Venkata Ramana Thanikella [2] and Isha Gupta [3]

1 Department of Management and Commerce, Amrita School of Arts and Sciences, Mysuru Campus, Amrita Vishwa Vidyapeetham, Bhogadi 570026, Karnataka, India
2 Birla Institute of Management Technology, Greater Noida 201306, Uttar Pradesh, India
3 Amity University, Noida 201303, Uttar Pradesh, India
* Correspondence: raghavendra.chandrika@gmail.com

**Abstract:** Climate change, global warming, and carbon emission are global issues. Countries are strengthening their environmental regulations to mitigate the emission problem. According to the pollution haven hypothesis, rich countries invest in emerging economies where the institutional framework is weak to migrate the emissions. With this background, this study examines the impact of the introduction of the carbon tax in India and environmental regulation restriction distance on India's inbound cross-border mergers and acquisitions (a form of foreign direct investment) volume using a 979 country-pair-year observation sample. The Tobit regression model findings suggest that carbon tax introduction and environmental regulation distance negatively impact India's inbound cross-border mergers and acquisitions volume. Furthermore, control of corruption intensifies its impact by effectively moderating them. The results indicate that India can avoid becoming a pollution haven by strengthening its environmental policies and controlling corruption. These results provide insight into strengthening the policies relating to environmental regulations and continuing the efforts required to control corruption in India.

**Keywords:** cross-border mergers and acquisitions; carbon tax; environmental regulation; control of corruption; India

**JEL Classification:** F2; O1; Q0; Q5

## 1. Introduction

Global warming has alarmed environmentalists and policymakers to tackle the greenhouse gas (GHG) emission issue, a significant contributor to environmental degradation. Carbon dioxide ($CO_2$) emission is responsible for GHG emissions (Olivier and Peters 2020). Even though world $CO_2$ emissions declined in 2020 by 5.8%, energy-related emissions stood at 31.5 Gt (International Energy Agency 2021). Around 2/3rd of the global emission is shared by emerging economies (EE) alone. The United Nations (UN) mentioned that the world must integrate and collaborate to achieve global sustainable development (UN 2019). It encourages the flow of Foreign Direct Investments (FDI) in EE to accomplish this. When FDI flow increases in EE, there will be a spillover effect on strengthening the institutional and regulatory framework governing environmental protection, and these economies will adopt their rules to match the international environmental protection standards (Abbas et al. 2021). Thus, exploring the role of the FDI flow of EE is important from an environmental perspective.

A significant portion of the FDI is cross-border mergers and acquisitions (CBM&A). CBM&A are the most preferred form of FDI. They have the edge over other forms such as greenfield investments and joint ventures, as multinational enterprises (MNEs) find it

advantageous due to time, cost, and resources (Slangen 2006). It is important to explore the CBM&A flow received by India because India has been the most attractive destination for MNEs. India's CBM&A has been dramatically increasing in the past three decades, reaching USD 27.211 million in 2019 (UNCTAD 2019, 2020). India's economic reforms, young population, skilled labor force, and vast market potential make it a top attractive destination for CBM&A activities. We argue that CBM&A should be explored from India's perspective because it is one of the most attractive destinations in the world, and moreover, its determinants are understudied to explain what and how various antecedents impact. With this background, we consider India's inbound CBM&A flow to understand how selected environmental factors impact.

Economic indicators (Erel et al. 2012; Ferreira and Massa 2010; Hyun and Kim 2010), formal (Buckley et al. 2016; Chari and Chang 2009) and informal (Ahern et al. 2012; Prasadh and Thenmozhi 2018) including institutional framework, geographical factors (di Giovanni 2005; Portes and Rey 2002), and resources (Deng and Yang 2015) are country-level factors that impact the CBM&A flow. Therefore, examining the country-level factors affecting CBM&A activities has fascinated researchers to understand the countries' characteristics influencing acquiring firms to choose a host country for their CBM&A activities.

In addition to various country-level factors mentioned above, environmental factors, including $CO_2$ emissions, play a vital role in CBM&A flows (Bose et al. 2021; Liu et al. 2021; Raghavendra et al. 2022). The Pollution Haven Hypothesis (PHH) argues that rich countries, to reduce their $CO_2$ emission, have been investing in EE where the institution is weak. When the home country's $CO_2$ emission is large, acquirers from such countries prefer and gain in CBM&A targeting firms in countries with low economic growth and weak environmental institutions (Bose et al. 2021). The acquirers gain in the short term when the host country's carbon emission is high, but in the long run, this gain fades (Liu et al. 2021). The home country's emission increases the likelihood of CBM&A (Bose et al. 2021). Therefore, $CO_2$ emissions play a vital role in CBM&A activities; hence, exploring whether the home country's $CO_2$ emissions influence the firms to undertake such activities in EE is fascinating. (Raghavendra et al. 2022) shows whether there are any symptoms of India being a pollution haven for MNEs entering through CBA activities and provides evidence that the home country's $CO_2$ emission is inversely related to India's inbound CBA volume. The government of India is very cautious and has been introducing various environmental protection policies and programs and strengthening its regulations.

Due to global awareness of climate change impacting human life, every country is preparing to war against environmental issues for sustainable development. India, the fastest-growing EE, is also trying hard to improve the quality of the environmental regulatory institution. Even though it is growing fast while being one of the world's top $CO_2$-emitting countries, it adopts environmental regulatory policies and programs. In 2010–11, the carbon tax was introduced due to a voluntary green initiative. India has been pricing 58.1% of the carbon emission from energy consumption since 2018, and the price has increased by EUR 14.33 since 2018 and reached EUR 14.43 in 2021 (OECD 2021). The carbon tax is an effective instrument as it helps minimize the abatement cost by encouraging firms to choose alternative, cleaner technologies (Das et al. 2021). However, the research on India's carbon tax is sparse, and we have less evidence of its effectiveness in shifting to cleaner technologies (Bhat and Mishra 2020). However, the policy has created the National Clean Energy and Environmental Fund due to the carbon tax, and this fund is utilized to fund cleaner technology-related research and development projects. This can negatively impact the acquiring firms entering India to migrate the carbon emissions. However, we do not know how carbon tax introduction affects the FDI or CBM&A activities.

Even though the government in every economy is introducing policies and strengthening regulations toward environmentally sustainable development, such environmental policies and regulation's effectiveness depends on the institutional setup. EEs either lack effective policies or/and lack an efficient regulatory environment (Dasgupta et al. 2002). Emerging host countries with high institutional distance increase uncertainty due to unfa-

miliar rules and policies (Contractor et al. 2014) and negatively impact CBM&A activities (Lahiri et al. 2013). However, how the environmental regulation distance between countries impacts the CBM&A activities is unclear. The evidence shows that non-compliance with environmental regulations will be costly for investors affecting operation costs and cash flows (Bose et al. 2021). However, weakness in the institutional setup provides an escape route to MNEs. Therefore, understanding the role of environmental regulation in CBM&A activities is essential, especially from an emerging host country's perspective.

Moreover, institutional quality impacts the success of any regulatory policies and programs. EE has poor institutional quality (Liou et al. 2016). One such indicator, corruption, is a worm-in-ointment and fails to maintain the strictness of the policies and programs in the system (Damania et al. 2003). Corruption can be both a boon and a bane for MNEs depending on the objective of internationalization (Hasan et al. 2017). It reduces the transparency in the system and hence becomes costly for the acquirers to operate in the corrupted host countries (Demirbag et al. 2007), especially when they intend to migrate the $CO_2$ emissions. However, with experience, they develop strategies to deal with it and create value through CBM&A activities (Barbopoulos et al. 2012). (Raghavendra et al. 2022) shows corruption control has benefited India from becoming a pollution haven from CBM&A activities. With the growing CBM&A activities and greater efforts to control corruption in India, we believe MNEs have gained experience and use corruption control measures to their advantage in the CBM&A activities. Therefore, it is imperative to understand the moderating role of corruption in sustainable development practices and CBM&A activities.

With this background, we propose to find answers to the following research questions: (1) Does the carbon tax introduction in India impact its inward CBM&A volume? (2) Does the environmental regulation distance between home and host countries impact India's inward CBM&A volume? and (3) Does India's corruption control, representing the formal institutional framework's quality, have a moderating role to play?

Home countries' MNEs must have been migrating to India to mitigate their carbon risk (Raghavendra et al. 2022). It is also seconded and supported by the fact that India has emerged as one of the top-emitting countries in the world during the last decades (Olivier and Peters 2020. Moreover, we argue that if MNEs consider India to migrate the carbon emissions, they expect similar environmental regulations compared to their home countries. Therefore, if the environmental regulation distance is larger, MNEs prefer to invest less in India as it brings uncertainties. Similarly, when India is trying to strengthen its environmental regulations, it should negatively impact the MNEs' investment decisions. Furthermore, we suspect control of corruption may intensify these relations. We contribute scientific value to the existing body of knowledge on the pollution haven hypothesis and show how India is a probable pollution haven for MNEs entering through the CBM&A route. Through our results, we add scientific value to the community, researchers, environmentalists, and regulatory bodies and show how India's control of corruption plays a vital role in CBM&A activities.

## 2. Review of Literature and Hypothesis Development

### 2.1. CBM&A Volume

The globalization of the economy and rapid industrialization are the root cause for dramatic growth in CBM&A activities. The driving forces of such CBM&A activities vary widely based on the business environment prevailing in home and host countries. Such forces are very complex as MNEs need to adapt and restructure their business operations to the new host country environment. Prolonged economic development in countries such as the United States of America has resulted in capital accumulation, and MNEs with huge capital started seeking complementary resources, markets, and technology, which started the CBM&A wave. This was mainly due to the globalization efforts of countries. Technological change, information technology in particular, has made it all possible to reduce cross-border transaction barriers. Globalization across the world has generated capital, goods, and services that are flowing from DE to EE. The phenomenon

of globalization has encouraged countries to sign various trade agreements. As a result, around 350 regional trade agreements were in force as per WTO as of March 2022. These agreements have been including new challenges to deal with, such as investor protection, labor protection, labor mobility, double taxation avoidance, and so on. Such agreements have been considered fruitful for CBM&A activities. The world's current situation of the COVID-19 pandemic and the Russia–Ukraine war has globally impacted CBM&A activities. CBM&A volume and the number of activities have dropped approximately by 20–50% worldwide during 2020 due to the pandemic. The strongly imposed sanctions by the world's leading powerful countries—such as the USA, European Union, Canada, Japan, Australia, and others, against Russia's war on Ukraine—have negatively impacted the CBM&A activities. In a nutshell, various national, international, and micro-level antecedents impact CBM&A activities worldwide. In this section, we enumerate such antecedents provided by empirical evidence from existing broad knowledge.

Various country-level antecedents impact the volume of CBM&A. The macroeconomic indicators, geopolitical factors, and cultural distance between acquiring and target nations are country-level factors that influence the CBM&A volume. Existing economic conditions, growth prospects, and the market size of the host country positively impact the flow of CBM&A volume (Guardo et al. 2013; Herger et al. 2008; Kiymaz 2004). The economy is more open to international trade and financial transactions and can attract more CBM&A flow (Hu et al. 2020). The restriction on the same discourages the flow as it negatively impacts the acquirer's gains (Moeller and Schlingemann 2005). As the economy grows, its financial market also develops and impacts the CBM&A volume (Ferreira and Massa 2010). The macro-economic indicators such as inflation rate (Boateng et al. 2017), interest rate, and money supply (Uddin and Boateng 2011) impact the CBM&A volume. Differential corporate tax encourages the MNEs to invest through the CBM&A route (McCann 2001). The tax treaties, bilateral agreements, and regional trade agreements guard against investment-related threats and double taxation; hence, positively impact the CBM&A volume (Hyun and Kim 2010). The weak exchange rate of the host against the home country encourages the MNEs to invest (Erel et al. 2012). The natural and technological resources (Deng and Yang 2015) and human resource development (Owen and Yawson 2010) positively impact CBM&A volume.

Apart from economic indicators, institutional framework and distance impact the CBM&A activities. The institutional quality (Owen and Yawson 2010) and regulatory strength (Buch and DeLong 2004) of the host country positively impact the CBM&A flow, while political instability (Basuil and Datta 2019) and corruption (Demirbag et al. 2007) discourages the MNEs to invest in host countries. In contrast, host countries with better minority investor protection attract more CBM&A flow (Choi et al. 2016), Quality of disclosure (Erel et al. 2012; Rossi and Volpin 2004) and implementation of IFRS (Francis et al. 2016) positively impact CBM&A volume.

Cultural distance is an informal institutional distance that negatively impacts the CBM&A volume (Ahern et al. 2012), while trading with cultural goods helps reduce this distance, hence, positively impacts (Li and Yang 2020). Lesser language barriers (Buckley et al. 2016) and psychic distance (Gulamhussen et al. 2016) help the MNEs to reduce the cultural distance and positively impact CBM&A volume. Even though religion is not found to have a consistent impact (Maung et al. 2020), religiosity and religious freedom in the country have a negative impact (Prasadh and Thenmozhi 2018). Cultural diversity is common when geographically distant countries, resulting in higher information and trade costs and negatively impacts the CBM&A volume (di Giovanni 2005; Gulamhussen et al. 2016; Portes and Rey 2002). This shows that research on country characteristics impacting the CBM&A volume is voluminous. However, the role of environmental degradation and MNEs' internationalization decision is still emerging and needs greater attention.

There is evidence to confirm environmental factors affecting the CBM&A activities (Bose et al. 2021; Liu et al. 2021; Raghavendra et al. 2022). The Pollution Haven Hypothesis (PHH) states that developed economies will invest in EE to reduce their home country

emissions and offshore them. These relocations of MNEs' business operations to EE where institutional weakness exists is not a globally sustainable development practice as such investments lead to increased emissions in host countries. Developed economies are reducing their emissions while EE is increasing (Olivier and Peters 2020). The investment flow across the nations through FDI impacting carbon emissions is controversial.

On the one hand, the pollution halo hypothesis argues that FDI can bring new technology and hence help reduce carbon emissions (Abid 2017; Balogh et al. 2017; Lee and Brahmasrene 2013). On the other hand, it is found that through FDI, MNEs try to offshore the emission, produce unfriendly environmental products in countries with weaker institutional setups to control environmental degradation, and import such products (Aller et al. 2021); and FDI increases the pollution in host countries (Huynh and Hoang 2019; Khan and Ozturk 2020; Mert and Caglar 2020; Shao 2018). When FDI inflow makes a host country a pollution haven, MNEs investing through CBM&A should also make the host country a pollution haven (Raghavendra et al. 2022). To support this argument, the existing evidence shows that the likelihood of CBM&A increases when the MNEs' home country emission is high. Such MNEs prefer an EE where the institutional setup is weak (Bose et al. 2021). Moreover, the acquirer gains in the short term when the host country's emission is higher (Liu et al. 2021). Therefore, exploring the environmental dimensions of CBM&A activities is a very important area of research.

### 2.2. Carbon Tax

The clean environment cess/carbon tax was introduced in 2010–11 due to a voluntary green initiative of India, which created the National Clean Energy and Environment Fund (NCEEF) through a Finance Bill in 2010–11. This is considered an essential milestone in the environmental policy reforms in India to reduce the carbon intensity of energy. India introduced a nationwide carbon tax on emissions caused by burning fossil fuels through this policy. NCEEF was introduced to contribute to the research and development (R&D) to develop technologies and clean environmental initiatives (Bhat and Mishra 2020). The carbon tax is the source of corpus for NCEEF. The carbon tax was a major environmental policy in India as this instrument can help minimize the abatement cost and switch towards cleaner technologies. Command-and-control regulation prevailing for a long time is ineffective in controlling firms using cost-inefficient abatement technologies. However, carbon tax holds such culprit firms liable for their emissions (Das et al. 2021).

To reduce carbon emissions to the global ideal level, countries must choose a carbon tax or introduce cap and trade policies. However, India has not introduced a full-fledged cap-and-trade scheme yet. Nevertheless, we believe introducing the carbon tax is an important step taken by the Indian government toward achieving its net-zero emission goal of 2070. As India is trying to improve the quality of environmental regulation, we believe that a carbon tax introduction can make India unattractive for MNEs looking to migrate the carbon emissions. In this background, we argue that introducing a carbon tax must negatively impact and reduce India's inbound CBM&A volume.

**Hypothesis 1a:** *India's introduction of carbon tax negatively impacts its inbound CBM&A volume.*

### 2.3. Environmental Regulation Distance

Examining environmental deterioration at various phases of economic development helps countries formulate environmental protection policies. The pollution heaven hypothesis argues that reducing the carbon footprint in high-income countries is possible when they offshore their emission to EE, where environmental regulation is weak (Aller et al. 2015). India has been taking various initiatives to improve environmental regulations. The host country's institutional quality (Owen and Yawson 2010) and regulatory strength (Buch and DeLong 2004) attract the CBM&A volume; however, institutional distance discourages it (Kedia and Bilgili 2015). Moreover, when the host country's environmental institution is weaker, it attracts MNEs and increases the CBM&A likelihood (Bose et al. 2021). Strengthen-

ing the environmental regulations by the host country can create uncertainty and increases the cost of operations and hence the cost of CBM&A. With this background, we expect environmental regulation distance to negatively impact the CBM&A volume as it causes uncertainty and the high cost of dealing with it. Therefore, when the environmental regulatory distance increases (decreases), India receives a lesser (larger) volume of CBM&A inflow.

**Hypothesis 2a:** *Environmental regulation distance negatively impacts India's inbound CBM&A volume.*

### 2.4. Moderating the Role of Corruption

When the $CO_2$ emission of the home country is high, MNEs from such countries prefer to invest through the CBM&A route in such host countries where environmental regulation is weak (Bose et al. 2021). India is the fastest-growing EE and strengthening environmental regulation. Nonetheless, EE's institutional setup is found to be more fragile compared to that of a DE (Liou et al. 2016). Corruption acts as a barricade while implementing any policies, including environmental policies and programs, as it reduces the grimness of such regulations (Damania et al. 2003). High corruption in the system can hamper the strictness of any regulations, including environmental rules and can increase $CO_2$ emissions (Fredriksson and Svensson 2003). At the same time, corruption reduces per capita income and decreases emission levels (Bae et al. 2017; Muhammad and Long 2021). However, when MNEs experience and develop strategies to deal with such high corruption, they gain from CBM&A activities (Barbopoulos et al. 2012).

Corruption prevailing in the system can make fail regulatory policies. The carbon tax is levied based on the marginal cost of reduction at a level of emissions against the standards, which are usually higher than the cost of complying with those standards. To effectively use this instrument, the government requires detailed estimates of reduction costs, and probable damage firms caused, and need to estimate the marginal cost of the same. Firms are demanded to provide data to the government authorities. With control of corruption, it becomes difficult for firms to bribe and hide the actual data on their emissions. MNEs who are environmentally conscious would prefer such host countries, and hence when corruption moderates, we suppose a positive impact of the carbon tax on CBM&A volume.

**Hypothesis 1b:** *With the existence of control of corruption, the introduction of the carbon tax has a positive effect on its inbound CBM&A volume.*

Furthermore, control of corruption in the host country intensifies this impact of environmental regulation distance on India's inbound CBM&A volume. As discussed before at the beginning of this Section 2.4, MNEs learn through their experience to develop strategies to overcome the uncertainty caused due to the changing environmental policies. When regulatory distance is high, we expect a smaller inflow of CBM&A in India. India's control of corruption will make it much costlier for such acquirers, and therefore, we expect the control of corruption in India to intensify the impact of environmental regulation distance and the CBM&A volume it receives.

**Hypothesis 2b:** *The control of corruption in India intensifies the impact of environmental regulation distance on its inbound CBM&A volume.*

## 3. Data and Methodology

### 3.1. Sample and Data

We designed our study to cover the period from 1990 to 2020 and gathered the CBM&A deal-level data from the Thomson Reuters Eikon database. We collected 5642 gross CBM&A deal data covering 81 acquiring countries targeting Indian firms. However, our research questions demand to have country-level data. Therefore, we converted the deal-level

data into 1537 country-pair-year observations following (Ahern et al. 2012; Prasadh and Thenmozhi 2018). By following the same authors, we defined our dependent variable as the cumulated dollar value of deals between acquirer/home and target/host in the acquisition year 't'. Then, we collected all our independent variables from various sources, and due to missing data, 979 observations were obtained finally to test the hypothesis.

Our independent variables include carbon tax introduction, environmental regulation, and control of corruption. Our first independent variable is the carbon tax dummy, '1' indicating post-2010, '0' otherwise. Furthermore, we use the environmental regulation distance between acquiring and target countries, defined by the environmental regulatory regime index. It consists of a combined score on regulatory stringency, structure, subsidies, and enforcement of environmental policies, indicating the quality of the environmental regulation of countries. A higher index represents better quality of regulation, and a lower score suggests poor quality of environmental regulation (Esty and Porter 2002). We use a moderating variable, the control of corruption index following (Liu et al. 2021; Luu et al. 2019; Raghavendra et al. 2022) from world governance indicators, varying from 0–100, with a lower value representing a high level of corruption and vice versa (Kaufmann et al. 2010).

We have included several control variables based on literature support, as explained in Section 2.1. We have used country-level characters, including the level of $CO_2$ emission in the target and home country measured in terms of kg/GDP sourced from the world bank (Raghavendra et al. 2022), financial development indicator, tax burden, an index from economic freedom indices from the Fraser Institute database following (Prasadh and Thenmozhi 2018), and financial depth of home country measured in terms of domestic credit to the private sector as a percentage of GDP collected from world bank following (Liang et al. 2018). Moreover, we have used specific distance measures, including economic distance measured in terms of differential GDP per capita between the acquiring and target country following (Ahern et al. 2012). The institutional and cultural distance was measured following Kogut and Sing's methodology using world governance indicators and Hofstede's cultural indices, respectively, following (Kogut and Singh 1988; Yoon et al. 2020). The geographical distance was measured as the distance between capital cities of the home and host countries following (Cuypers and Ertug 2015; Jongwanich et al. 2013). Efficiency distance was measured as differential wages in the acquiring and target countries, and data was gathered from the world bank following (Dikova et al. 2019).

### 3.2. Methodology

We proposed to examine our hypothesis empirically using the following model:

Ln (CBM&A Volume $_{ij,\,t}$) = β1 (CT$_j$) + β2 (ERR_distance$_{ij}$) + β3 (CoC$_j$) + β4 (CT$_j$*CoC$_j$) +

β5 (ERR_distance$_{ij}$*CoC$_j$) + β6 (Controls) + Acquiring country dummies + Year dummies + Constants + μ$_{ij,t}$    (Model 1)

where *i* indicates acquiring country; *j* indicates the target country; *t* indicates the year of acquisition. CT represents the carbon tax dummy, ERR represents the environmental regulation restriction distance measure, and CoC represents the control of corruption index.

When we analyzed the nature of our dependent variable, we understood it is left-censored data as no deals occurred during a *t* time between *i* and *j*. We planned to use the Tobit model to address censoring issues arising from zero values of CBM&A (Ahern et al. 2012; Prasadh and Thenmozhi 2018; Raghavendra et al. 2022). When there is no value of CBM&A between acquiring countries and India, it results in a zero-value issue in the estimated volume, which is a left censored case. The Tobit regression can be defined as

$$Y_j = \max\left(Y_j^*, 0\right) \quad (1)$$

The $Y_j^*$ of Equation (1) can be found by applying the classical linear regression equation as given below

$$Y_j^* = \beta X_j + \varepsilon_j \quad (2)$$

Moreover, a value 0 on the dependent variable mean truly a value 0 or the data i censored, or it can be $Y_j = Y_j^*$. Therefore,

$$Y_j = \begin{cases} 0, & Y_j^* \leq 0 \\ Y_j^*, & 0 < Y_j < 1 \\ 1, & Y_j \geq 1 \end{cases} \tag{3}$$

With this background, Model 1 was developed, and STATA statistical package was used to analyze the data. We also have specific time-invariant variables, such as cultural distance measures; to address this issue, we introduced year-fixed effects and acquired country-fixed effects to capture the time-invariant variables and time shocks. We have taken one-year lag values of all independent variables to avoid the endogeneity issues with the dependent variable, as suggested by (Dikova et al. 2019). We have tested for multicollinearity issues using VIF, and all the variables have a value of less than 10. This gave us the confidence to test the proposed model by applying Tobit regression. We also planned to use alternative specifications of the models to check for robustness and find similar results.

## 4. Results and Discussion

### 4.1. Summary Statistics

Table 1 shows the summary statistics of all variables with 979 observations. The volume of CBM&A has a mean value of USD 1.751 million, with a standard deviation of 2.554, a minimum being 0 and a maximum of 8.760. The carbon tax (CT) introduced in India is a dummy variable with a mean of 0.421 and a standard deviation of 0.494. The environment regulation restriction (ERR) score representing environmental regulation distance has a mean score of 1.16 with a standard deviation of 0.817. The regulatory distance was found to be both a positive distance indicating acquiring firms are from countries *i* with stronger environmental regulations, and a negative score indicates that acquiring firms are from countries with weaker environmental regulations. As shown in the table, the minimum ERR score is −0.177, indicating countries with weaker regulation, and the maximum is 3.062, indicating countries with stronger regulation than country *j*. The mean ERR score is 1.16, with a standard deviation of 0.817. The corruption control index of country j has a mean value of 42.589 with a standard deviation of 4.251. The minimum value of the index is 35.545, and the maximum is 49.519. The score varies from 0 to 100; a higher value indicates better control of corruption. India's corruption index indicates the efforts of the government to control it.

**Table 1.** Summary statistics of all variables.

| Variable | Obs | Mean | Std. Dev. | Min | Max |
|---|---|---|---|---|---|
| Volume_CBM&A | 979 | 1.751 | 2.554 | 0.000 | 8.760 |
| CT_T | 979 | 0.421 | 0.494 | 0.000 | 1.000 |
| ERR Distance | 979 | 1.160 | 0.817 | −0.177 | 3.062 |
| CoC_T | 979 | 42.589 | 4.251 | 35.545 | 49.519 |
| $CO_2$_A | 979 | 0.485 | 0.365 | 0.070 | 1.525 |
| $CO_2$_T | 979 | 1.081 | 0.074 | 0.940 | 1.232 |
| Taxburden_T | 979 | 75.533 | 3.657 | 46.800 | 79.400 |
| FinDepth_A | 979 | 80.274 | 44.116 | 13.331 | 190.674 |
| Ex. Rate_growth | 979 | −0.006 | 0.068 | −0.209 | 0.161 |
| Eco_Distance(GDPpc) | 979 | 2.834 | 1.221 | 0.000 | 4.793 |
| Inst_Distance | 979 | 0.003 | 0.002 | 0.000 | 0.009 |
| Cul_Distance | 979 | 0.002 | 0.001 | 0.000 | 0.004 |
| Geo_Distance | 979 | 8.362 | 1.377 | 0.000 | 9.594 |
| Eff_Distance | 979 | 60.346 | 15.846 | 23.470 | 81.790 |

Figure 1 shows that more CBM&A flow is observed from high-income group countries with USD 213,317.22, lower-income group countries with USD 48,443, and middle-income group countries that have invested the USD 13468 value through CBM&A in India. Our sample comprises around 77% of the CBM&A flow from high-income countries.

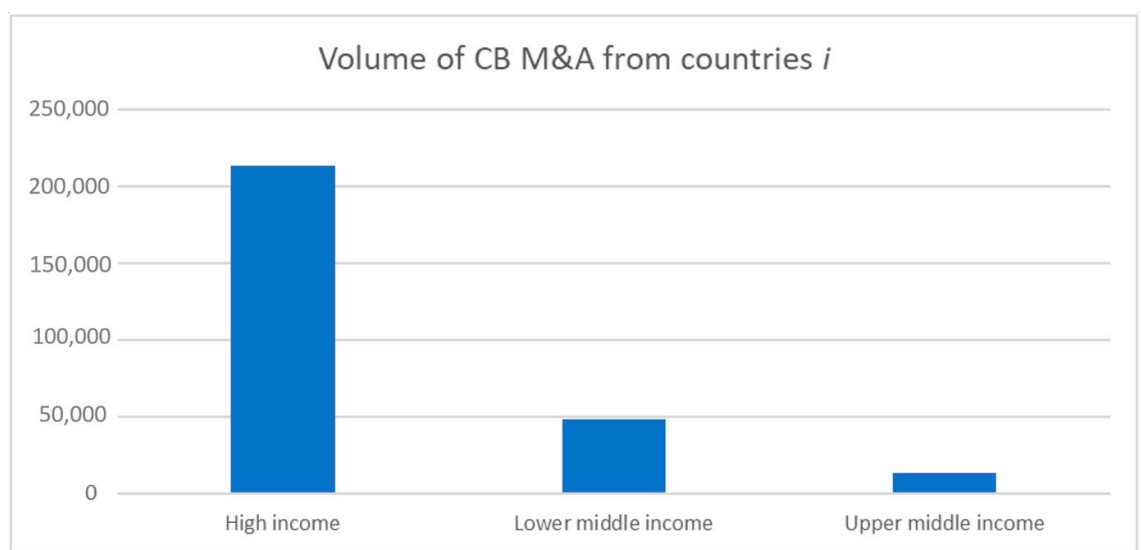

**Figure 1.** Sum of CBM&A volume from countries *i*, classified based on income groups. (Source: Author's consolidation based on World Bank's country classification).

Table 2 shows the correlation matrix of all variables. None of the variables are highly correlated, having coefficients of more than 0.8. We also tested VIF for multicollinearity issues and found the VIF value was less than 10 for all the model variables, as shown in Table 3. Hence, there was no issue of multicollinearity in the models.

**Table 2.** Correlation matrix of all variables.

| | 1 | 2 | 3 | 4 | 5 | 6 | 7 | 8 | 9 | 10 | 11 | 12 | 13 | 14 |
|---|---|---|---|---|---|---|---|---|---|---|---|---|---|---|
| Volume CBM&A (1) | 1 | | | | | | | | | | | | | |
| $CO_2\_A$ (2) | −0.147 | 1 | | | | | | | | | | | | |
| CT_T (3) | 0.094 | −0.110 | 1 | | | | | | | | | | | |
| ERR Distance (4) | 0.327 | −0.576 | 0.000 | 1 | | | | | | | | | | |
| CoC_T (5) | −0.004 | 0.001 | −0.074 | −0.019 | 1 | | | | | | | | | |
| $CO_2\_T$ (6) | −0.160 | 0.119 | −0.495 | −0.024 | −0.364 | 1 | | | | | | | | |
| Taxburden_T (7) | 0.159 | −0.092 | 0.408 | 0.026 | 0.041 | −0.589 | 1 | | | | | | | |
| FinDepth_A (8) | 0.205 | −0.376 | 0.117 | 0.442 | −0.045 | −0.134 | 0.109 | 1 | | | | | | |
| Ex. Rate_growth (9) | −0.018 | 0.043 | −0.050 | 0.008 | 0.076 | 0.025 | −0.029 | −0.024 | 1 | | | | | |
| Eco_Distance (10) | 0.148 | −0.585 | −0.139 | 0.714 | −0.062 | 0.086 | −0.048 | 0.428 | −0.046 | 1 | | | | |
| Inst_Distance (11) | 0.004 | −0.054 | 0.089 | −0.140 | 0.055 | −0.204 | 0.144 | −0.146 | 0.034 | −0.283 | 1 | | | |
| Cul_Distance (12) | 0.074 | −0.047 | −0.003 | 0.181 | 0.002 | 0.007 | −0.005 | 0.152 | −0.011 | −0.017 | 0.101 | 1 | | |
| Geo_Distance (13) | −0.213 | −0.353 | 0.004 | 0.309 | −0.008 | −0.016 | 0.013 | 0.180 | −0.020 | 0.414 | 0.229 | 0.254 | 1 | |
| Eff_Distance (14) | 0.114 | −0.315 | −0.033 | 0.546 | −0.060 | −0.012 | 0.020 | 0.200 | 0.009 | 0.818 | −0.266 | −0.089 | 0.356 | 1 |

**Table 3.** VIF values for checking multicollinearity among the variables in all the models.

| Variable | VIF | |
| --- | --- | --- |
| | Model 1 | Model 2 |
| CT_T | | 1.63 |
| ERR Distance | | 2.50 |
| CoC_T | 1.26 | 1.37 |
| $CO_2$_A | 2.01 | 2.22 |
| $CO_2$_T | 2.05 | 2.45 |
| Taxburden_T | 1.62 | 1.64 |
| FinDepth_A | 1.51 | 1.55 |
| Ex. Rate_growth | 1.02 | 1.03 |
| Eco_Distance(GDPpc) | 4.50 | 7.93 |
| Inst_Distance | 1.38 | 1.45 |
| Cul_Distance | 1.14 | 1.23 |
| Geo_Distance | 1.56 | 1.60 |
| Eff_Distance | 3.65 | 4.34 |
| Mean VIF | 1.97 | 2.38 |

*4.2. Impact of the Introduction of the Carbon Tax and Environmental Regulation Distance on CBM&A Volume*

Table 4 provides the results of the Tobit regression model showing the impact of the carbon tax introduced in India and the environmental regulation distance between India and acquiring countries, on the CBM&A volume with 979 country-pair-year observations. Model 1 examines the impact of control and moderating variables on the CBM&A volume. We found that control of corruption in the target country had a negative and statistically significant ($p < 0.1$; β = −3.968) impact on the CBM&A volume of India. It indicates that when the target country *j* controlled corruption, its volume of CBM&A decreased by 3.97%. The negative impact can be attributed to uncertainty caused by efforts to improve institutional quality and the high cost of bribing due to corruption control in the target country. Moreover, institutional distance ($p > 0.1$; β = −36.016), financial depth ($p > 0.1$; β = −0.0001), exchange rate growth ($p > 0.1$; β = −1.662), $CO_2$ emission of home country, and $CO_2$ emission of target country ($p > 0.1$; β = −32.192) had a negative impact on the CBM&A volume but statistically was not significant. Cultural distance ($p < 0.1$; β = −18252.9) was found to have a negative impact with a very high coefficient and was statistically significant. The tax burden of the target country ($p < 0.1$; β = 1.703), representing the financial development, was found to have a significant positive impact on the CBM&A volume. Economic distance ($p < 0.05$; β = 1.472), geographical distance indicating the cost of information and trade ($p < 0.01$; β = 19.66), and efficiency distance ($p < 0.05$; β = 0.157) had a positive and statistically significant impact on the volume of CBM&A.

Model 2 includes the major independent variables along with control variables. Control variables had a similar impact, except that the $CO_2$ emission of the target country became statistically significant with a high coefficient ($p < 0.1$; β = −778.213), indicating that when the target country's $CO_2$ emission increased, the volume increases of CBM&A decreased by 778.213%. It shows that MNEs preferred targets to be located in less emitting economies. The $CO_2$ emission of the home countries was found to impact the CBM&A volume negatively and was statistically significant ($p < 0.01$; β = −7.098). High-emitting countries invested less in Indian targets. In other words, when the $CO_2$ emission of the acquiring country was high (low), India received 7.098% less (more) volume of CBM&A from such countries. India's carbon tax introduction had a significant negative impact ($p < 0.1$; β = −170.661), meaning India received 170.661% less CBM&A volume from home countries after the introduction and found support for our hypothesis (Hypothesis 1a). The environment regulation restriction distance between the acquiring and target country was found to have a negative and significant ($p < 0.05$; β = −27.339) impact on the CBM&A volume *(Hypothesis 2a)*. This shows that when the regulation restriction distance was large (less), there was a 27.339% reduction (increase) in the CBM&A volume in India.

**Table 4.** Tobit Model results for the impact of the carbon tax and environmental regulation restriction distance on CBM&A volume and moderating effect of control of corruption of target country on the impact of independent variables on the CBM&A volume. The models include control variables showing an impact on CBM&A volume.

| | Model 1 | | Model 2 | | Model 3 | |
|---|---|---|---|---|---|---|
| | Coeff. | *p*-Value | Coeff. | *p*-Value | Coeff. | *p*-Value |
| Independent variables | | | | | | |
| CT_T | | | −170.661 * | 0.066 | 1124.464 * | 0.080 |
| ERR Distance | | | −27.339 ** | 0.017 | −27.855 ** | 0.016 |
| Moderating variable | | | | | | |
| CoC_T | −3.968 * | 0.093 | −9.139 * | 0.074 | 20.914 * | 0.081 |
| Interaction effect | | | | | | |
| CT_T*CoC_T | | | | | −30.047 * | 0.078 |
| ERR Distance*CoC_T | | | | | 0.048 | 0.242 |
| Control Variables | | | | | | |
| $CO_2$_T | −32.192 | 0.119 | −778.213 * | 0.060 | −766.747 * | 0.062 |
| $CO_2$_A | −7.098 *** | 0.002 | −7.098 *** | 0.002 | −15.238 *** | 0.001 |
| Taxburden_T | 1.703 * | 0.076 | 1.689 * | 0.073 | 1.666 * | 0.074 |
| FinDepth_A | −0.0001 | 0.990 | −0.001 | 0.916 | −0.002 | 0.750 |
| Ex. Rate_growth | −1.662 | 0.421 | −2.280 | 0.270 | −2.072 | 0.317 |
| Eco_Distance(GDPpc) | 1.472 ** | 0.035 | 0.769 | 0.293 | 0.920 | 0.215 |
| Inst_Distance | −36.016 | 0.692 | −36.774 | 0.684 | −40.148 | 0.661 |
| Cul_Distance | −18,252.09 *** | 0.000 | −30,418.82 *** | 0.003 | −29,443.08 *** | 0.005 |
| Geo_Distance | 19.660 *** | 0.004 | 34.773 *** | 0.006 | 33.286 *** | 0.009 |
| Eff_Distance | 0.157 ** | 0.026 | 0.169 ** | 0.016 | 0.167 ** | 0.019 |
| Acquiring country FE | Yes | | Yes | | Yes | |
| Year FE | Yes | | Yes | | Yes | |
| No. Obs. | 979 | | 979 | | 979 | |
| LR chi2 | 799.050 *** | 0.000 | 808.78 *** | 0.000 | 812.79 *** | 0.000 |
| Pseudo R2 | 0.251 | | 0.254 | | 0.256 | |
| Log-likelihood | −1190.805 | | −1185.940 | | −1183.931 | |
| Constant | −61.706 | 0.460 | 1007.932 | 0.133 | −288.887 ** | 0.021 |

Note: *, **, *** represent significance at 10%, 5% and 1% levels, respectively.

### 4.3. Moderation of Target Country's Control of Corruption

Model 3, in Table 4, shows the moderating effect of target country corruption control with major independent and control variables. We found a similar impact of control variables on CBM&A volume in models 1 and 2. We brought the moderation effect in the form of interaction terms in the model to examine the same impact. We found that the interaction effect of target country corruption with carbon tax positively impacted CBM&A volume (Hypothesis 1b). The results show that with the existing level of corruption control in the target country, the introduction of the carbon tax has impacted the CBM&A volume positively and was statistically significant ($p < 0.1$). This interaction resulted in a 1094.417% (1124.464–30.0467) increase in the CBM&A volume in India. Furthermore, when we examined the interaction effect of target country corruption control with ERR distance, we found no substantial difference in the coefficient, and the interaction term is not statistically significant ($p > 0.1$). Hence, we have not found support for our hypothesis (Hypothesis 2b).

We checked the robustness of the models by taking the alternative specifications and finding similar evidence. First, we introduced emission differential values to the model and found a similar negative significant impact ($p < 0.01$). Moreover, we replaced the economic distance measure with the GDP of the host country and found identical effective results. Furthermore, when we dropped control variables, we found that carbon tax and environmental regulation distance had a significant positive effect ($p < 0.01$). We also modified the model by introducing gravity model variables (Ahern et al. 2012; Prasadh and Thenmozhi 2018), and we found that carbon tax introduction in the host country

had a negative impact on CBM&A volume while environmental regulation distance had a positive impact. Due to word count limitations, we have not brought these results to this paper.

## 5. Discussion

In this study, we tested the impact of the introduction of the carbon tax and environmental regulation distance on the inbound CBM&A volume India receives. We used 979 country-pair-year observations from 1990 to 2020 by applying the Tobit regression model to test our hypothesis. We used country-fixed effects and time dummies to capture the time-invariant variables and time shocks in the model, performed a multicollinearity test to confirm no highly related variables, and then ran the Tobit regression model. The design we developed to perform the data analyses was similar to various other authors who used volume as their dependent variable. Identical to (Ahern et al. 2012; Prasadh and Thenmozhi 2018) and others, our dependent variable was censored at the left due to the zero-values issue observed in it; hence, the Tobit regression has been applied following those authors.

The environmental Kuznets curve hypothesis argues that the rich countries emit less while EE emits more $CO_2$. The rich countries that emit less $CO_2$ due to strict regulations and public awareness invest and relocate their business operations to other EE where the institution is weak as per the pollution haven hypothesis. Moreover, according to the pollution haven hypothesis, they prefer EE, where institutional weakness exists while investing abroad to reduce their emission and migrate them. With this background, we employed environmental regulation distance to analyze its impact on the CBM&A volume that India receives. In line with the findings of (Bose et al. 2021), we found that MNEs preferred to invest in India with a large volume when the environmental regulation distance was smaller, which means MNEs expect a similar regulatory environment in India to increase their volume of CBM&A flow. This is evident from a similar finding (Kedia and Bilgili 2015; Lahiri et al. 2013) that when institutional distance is large, it discourages the MNEs from investing in such host countries as it results in uncertainty in policies and programs (Contractor et al. 2014). Therefore, India should further strengthen its environmental regulations to attract more CBM&A volume.

Furthermore, we tested the introduction of the carbon tax policy in India and its impact on its inbound CBM&A volume. We found that introduction of the policy had a negative impact on the CBM&A volume. Our results are similar to the results of (Wall et al. 2019), who show that introducing fiscal policy carbon instruments such as carbon tax attracts green FDI. In other words, when the CBM&A flow is motivated to migrate the carbon emission, India's carbon pricing/tax policy has effectively created a barricade. This indicates carbon tax is an effective policy for controlling home countries that intend to invest through CBM&A to migrate their $CO_2$ emissions.

We also examined the moderating role of control of corruption in the host country, India, to analyze how it plays its part. We found that control of corruption negatively impacted the CBM&A flow. Our results contrasted with those of (Candau and Dienesch 2017; Wang et al. 2020), who showed the role of corruption in making host countries a pollution haven. Moreover, without the interaction of the target country's corruption control, the carbon tax introduction was negative as it attracted the carbon cess/tax. Moreover, with the interaction of control of corruption, we found that carbon tax introduction positively impacted CBM&A volume.

To this end, we contribute to the existing literature on environmental policies and sustainable development goals by adding knowledge from a cross-border mergers and acquisitions perspective. Our results are unique as we are among the very few who have shown environmental policies impact CBM&A activities. We add scientific value to the scholars to further explore whether India benefits from such CBM&A activities through the pollution halo effect or becoming a complete victim of pollution haven. Based on this and needed future studies, policymakers and regulatory bodies need to strengthen the environmental policies in India to attract green CBM&A volume. They also need to keep

their efforts to control corruption and keep the system clean to avoid MNEs taking undue advantage of the institutional setup's weaknesses.

## 6. Conclusions

This study examines the impact of the introduction of the carbon tax in India and environmental regulation restriction distance on India's cross-border mergers and acquisitions volume. Furthermore, we examine the moderating role of control of corruption. The study suggests that carbon tax introduction and environmental regulation distance negatively impact India's inbound cross-border mergers and acquisitions volume. The moderating role of control of corruption intensifies the impact of home countries' carbon emissions. These results show that India can avoid becoming a pollution haven by strengthening its environmental policies and institutional setup.

These results provide insight into strengthening the policies relating to environmental regulations and controlling corruption in India. India's environmental policies and programs must be strengthened to reduce its carbon footprint. At the same time, India should continue to support environmental regulations to meet global standards and stand on the frontline to fight against global warming. We suggest that policymakers, economists, and environmentalists work together to achieve India's net-zero target by 2070 and reach sustainable development goals. Our study does not consider the firm-level data; hence, we recommend that scholars consider and explore how firm-level emissions impact their CBM&A decisions. Moreover, we have not considered data on the carbon tax introduction in home countries. Further studies should include such data to understand how MNEs behave for such policies. Furthermore, future research needs to be done to understand whether fewer emitting countries are using this as an offshore strategy as explained by the pollution heaven hypothesis, examining the impact of CBM&A volume on India's $CO_2$ emission and environmentally unfriendly products production and exports.

**Author Contributions:** Conceptualization, C.R., M.R.; methodology, C.R.; software: C.R.; validation: C.R.; formal analysis, C.R., M.R.; investigation, C.R., M.R.; resources, C.R., M.R., I.G.; data curation, C.R., M.R.; writing—original draft preparation, C.R.; writing—review and editing, M.R., V.R.T.; visualization, C.R., M.R., V.R.T.; supervision, M.R., V.R.T.; project administration, C.R., M.R., V.R.T., I.G.; funding acquisition, C.R., M.R., V.R.T., I.G. All authors have read and agreed to the published version of the manuscript.

**Funding:** The ACP was funded partially by Birla Institute of Management Technology.

**Data Availability Statement:** Data can be provided on demand.

**Conflicts of Interest:** The authors declare no conflict of interest.

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
