# Peer review of "Impact of Carbon Tax and Environmental Regulation on Inbound Cross-Border Mergers and Acquisitions Volume: An Evidence from India"

_ijfs, doi:10.3390/ijfs10040106_

Round 1

Reviewer 1 Report

Comments:

1. This study aims at testing the polluting haven hypothesis. However, the analysis is at the country-pair-year level. To properly test this hypothesis, we need to have firm-level or at least industry-level information. The reason is that firms from high-emitting countries may not operate in pollution industries. As a result, there is no incentive for pollution migration or M&A.

Therefore, I suggest author(s) use more detailed data to answer this question.

2. Some testing results are not consistent with the pollution haven hypothesis. For instance, the high-emitting countries have fewer M&As with Indian firms.

3. Also, author(s) need to collect information on carbon tax policies in the acquiring countries as well.

4. Some writing is unclear. For instance, it is stated that "Our independent variables include Our seond independent ..." The sentence is incomplete and may miss important information.

Author Response

Response to Reviewer’s Comments

Title: Impact of Carbon Tax and Environmental Regulation on Inbound Cross-Border Mergers and Acquisitions Volume: An Evidence from India

Dear Professor and Reviewers,

We would like to thank you for considering reviewing our article and providing valuable suggestions to improve the quality. We have considered and made some major revisions to the manuscript based on the suggestions. Please find the changes done through the track change option in MS word. And you can find the detailed response to each of the reviewers’ comments here below.

#REVIEWER 1

  1. This study aims at testing the polluting haven hypothesis. However, the analysis is at the country-pair-year level. To properly test this hypothesis, we need to have firm-level or at least industry-level information. The reason is that firms from high-emitting countries may not operate in pollution industries. As a result, there is no incentive for pollution migration or M&A. Therefore, I suggest author(s) use more detailed data to answer this question.

Reply: We appreciate the reviewer’s point of view here. And to your kind notice, we would like to inform you that we have used firm-level data collected from the Thomson Reuters EIKON database. Our objective was to analyze how the introduction of the carbon tax and environmental regulation impact the VOLUME of cross-border mergers and acquisitions (CBM&A). The volume is the sum of the dollar value of CBM&A flow between the acquiring country and the target nation. Since the volume is country-level data, we had to convert the deal/firm-level data into country-level data. Please consider reading the literature ( Ahern et al., 2012; Prasadh & Thenmozhi, 2018; Raghavendra et al., 2022). We have used similar measurement tools as these authors have used. And hence we request that you consider the country-pair-year observations measured from the firm-level deal data.

  1. Some testing results are not consistent with the pollution haven hypothesis. For instance, the high-emitting countries have fewer M&As with Indian firms.

Reply: This output shows that less emitting countries are investing more, which indicates that developed economies that have reduced their emissions have been investing more; on the other hand, more emitting countries are investing less in Indian firms. This is because high-emitting countries are emerging economies, and as per the EKC hypothesis, while they grow, they emit more. They internationalise their operations through FDI and its forms only when they become rich/developed country and reduce their emission. Therefore, CO2 emission has a negative impact on CBM&A flow in India. This is supported by the pollution haven hypothesis and EKC hypothesis.

  1. Also, the author(s) need to collect information on carbon tax policies in the acquiring countries as well.

Reply: We thank the reviewer for suggesting us a new point of view to consider. However, our objective was to analyse India’s policy change’s impact on introducing a carbon tax in the system. And hence, we didn’t collect this data on the carbon tax policy of those home countries as we did not think it was important. Moreover, we would like to inform you that we have used 81 acquiring country’s firms’ data in this study. At this stage of revision with time and resource constraints, if you suggest including it, it is practically not possible to collect all the home countries’ carbon tax data. We realized this is one of the limitations of our study, and hence we have included it under limitations. Please refer to lines 500-502 for changes done.

  1. Some writing is unclear. For instance, it is stated that "Our independent variables include Our second independent ..." The sentence is incomplete and may miss important information.

Reply: We apologize for such a mistake. We have made corrections in this regard. Please refer to lines 277-278 for the changes done.

Reviewer 2 Report

I assess the article entitled Impact of Carbon Tax and Environmental Regulation on Inbound Cross-Border Mergers and Acquisitions Volume: Evidence from India positively. In my opinion, the subject is very important from the point of view of the state, enterprises, and society. The whole world is struggling with the reduction of green gas emissions and with the initiative of sustainable development. The authors noted the problem According to the pollution haven hypothesis, rich countries invest in emerging economies where the institutional framework is weak to migrate the emissions. In this background, this study examines the impact of the introduction of the carbon tax in India and environmental regulation restriction distance on India's inbound cross-border mergers and acquisitions (a form of foreign direct investment) volume using a 979 country-pair-year observation sample. The Tobit regression model was used in the study. It has been well used, but one of the shortcomings is the lack of a detailed description of the research tool used in the Methodology. This should be completed. Another issue that requires comment is the description of cross-border mergers and acquisitions (CBM & A), why did the authors focus on this type of FDI? why do they have an advantage over other forms of FDI? what makes them stand out? what characterizes them? This is important as it is the background to all the work. They conclude the research questions posed with the hypotheses. The hypotheses have been correctly formulated, but only in the opinion of the reviewer should be detailed Hypothesis1b: With the existence of control of corruption, the introduction of the carbon tax has a positive impact on its inbound CBM & A volume. The impact is understood in various ways, it can be economic, social, environmental, etc. I propose to change the name of hypothesis 1b to: With the existence of control of corruption, the introduction of the carbon has a positive effect on its inbound CBM & A volume. With this hypothesis, the authors should explain how they limited the impact of other factors that also affect the CBM & A volume. I am assessing it positively with other research. in 2.1. authors can add information about globalization, preferential trade agreements and the current world situation (COVID-19, war in Ukraine). The article should emphasize the fact that the pollution halo hypothesis argues that FDI can bring new technology and hence help reduce carbon emissions. Because FDI is not always associated with an increase in environmental pollution (unfortunately, in most cases, yes). The above-mentioned comments are of a discussion nature and in no way diminish the value of the work. I believe that there is a research gap in this area and that the article raises very important issues. There is a lack of research on the environmental dimensions of CBM & A activities, therefore it is an important issue.

Author Response

Response to Reviewer’s Comments

Title: Impact of Carbon Tax and Environmental Regulation on Inbound Cross-Border Mergers and Acquisitions Volume: An Evidence from India

Dear Professor and Reviewer,

We would like to thank you for considering reviewing our article and providing valuable suggestions to improve the quality. We have considered and made some major revisions to the manuscript based on the suggestions. Please find the changes done through the track change option in MS word. And you can find the detailed response to each of the reviewers’ comments here below.

#REVIEWER 2

I assess the article entitled Impact of Carbon Tax and Environmental Regulation on Inbound Cross-Border Mergers and Acquisitions Volume: Evidence from India positively. In my opinion, the subject is very important from the point of view of the state, enterprises, and society. The whole world is struggling with the reduction of green gas emissions and with the initiative of sustainable development. The authors noted the problem. According to the pollution haven hypothesis, rich countries invest in emerging economies where the institutional framework is weak to migrate the emissions. In this background, this study examines the impact of the introduction of the carbon tax in India and environmental regulation restriction distance on India's inbound cross-border mergers and acquisitions (a form of foreign direct investment) volume using a 979 country-pair-year observation sample.

  1. The Tobit regression model was used in the study. It has been well used, but one of the shortcomings is the lack of a detailed description of the research tool used in the Methodology. This should be completed.

Reply: We have included the description of the research tool in detail to include Data collection, Data processing, and Analysis. However, as you have mentioned in this comment, we missed a briefing on statistical tool derivation, so we have included it in this revised version. Please refer to lines 318 to 329 for the same.

  1. Another issue that requires comment is the description of cross-border mergers and acquisitions (CBM & A), why did the authors focus on this type of FDI? why do they have an advantage over other forms of FDI? what makes them stand out? what characterizes them? This is important as it is the background to all the work.

Reply: To address the points mentioned in this suggestion, we have incorporated the changes in the manuscript to add answers to questions. Please refer to lines 42 to 51 for the same.

  1. They conclude the research questions posed with the hypotheses. The hypotheses have been correctly formulated, but only in the opinion of the reviewer should be detailed Hypothesis1b: With the existence of control of corruption, the introduction of the carbon tax has a positive impact on its inbound CBM & A volume. The impact is understood in various ways, it can be economic, social, environmental, etc. I propose to change the name of hypothesis 1b to: With the existence of control of corruption, the introduction of carbon has a positive effect on its inbound CBM & A volume. With this hypothesis, the authors should explain how they limited the impact of other factors that also affect the CBM & A volume. I am assessing it positively with other research.

Reply: We have changed Hypothesis 1b as suggested. Please refer to lines 264 to 266. We have used multiple control variables to limit the impact of other factors on CBM&A volume. And that has been enumerated in the methods and in results sections.

  1. In 2.1. authors can add information about globalization, preferential trade agreements and the current world situation (COVID-19, war in Ukraine). The article should emphasize the fact that the pollution halo hypothesis argues that FDI can bring new technology and hence help reduce carbon emissions. Because FDI is not always associated with an increase in environmental pollution (unfortunately, in most cases, yes).

Reply: We have added a paragraph in section 2.1 to include information about the points mentioned in the suggestion. Please refer to lines 141 to 164 for the changes done.

The above-mentioned comments are of a discussion nature and in no way diminish the value of the work. I believe that there is a research gap in this area and that the article raises very important issues. There is a lack of research on the environmental dimensions of CBM & A activities. Therefore, it is an important issue.

Round 2

Reviewer 1 Report

Good revision.